# Exploring Lysophosphatidylcholine as a Biomarker in Ischemic Stroke: The Plasma–Brain Disjunction

**DOI:** 10.3390/ijms251910649

**Published:** 2024-10-03

**Authors:** Justin Turpin, Steven Wadolowski, Willians Tambo, Daniel Kim, Yousef Al Abed, Daniel M. Sciubba, Lance B. Becker, David Ledoux, Junhwan Kim, Keren Powell, Chunyan Li

**Affiliations:** 1Department of Neurosurgery, Zucker School of Medicine at Hofstra/Northwell, Hempstead, NY 11549, USA; 2Translational Brain Research Laboratory, Feinstein Institutes for Medical Research, Manhasset, New York, NY 11030, USA; 3Institute of Bioelectronic Medicine, Feinstein Institutes for Medical Research, Manhasset, New York, NY 11030, USA; 4Elmezzi Graduate School of Molecular Medicine at Northwell Health, Manhasset, New York, NY 11030, USA; 5Biology Department, Boston College, Chestnut Hill, MA 02467, USA; 6Donald and Barbara Zucker School of Medicine at Hofstra/Northwell, Hempstead, NY 11549, USA; 7Laboratory for Critical Care Physiology, Feinstein Institutes for Medical Research, Manhasset, New York, NY 11030, USA

**Keywords:** lysophosphatidylcholine, ischemic stroke, biomarker, prognostic marker, lipid, transient middle cerebral artery occlusion, LPC(18:1)

## Abstract

Lipids and their bioactive metabolites, notably lysophosphatidylcholine (LPC), are increasingly important in ischemic stroke research. Reduced plasma LPC levels have been linked to stroke occurrence and poor outcomes, positioning LPC as a potential prognostic or diagnostic marker. Nonetheless, the connection between plasma LPC levels and stroke severity remains unclear. This study aimed to elucidate this relationship by examining plasma LPC levels in conjunction with brain LPC levels to provide a deeper understanding of the underlying mechanisms. Adult male Sprague–Dawley rats underwent transient middle cerebral artery occlusion and were randomly assigned to different groups (sham-operated, vehicle, LPC supplementation, or LPC inhibition). We measured multiple LPC species in the plasma and brain, alongside assessing sensorimotor dysfunction, cerebral perfusion, lesion volume, and markers of BBB damage, inflammation, apoptosis, and oxidative stress. Among five LPC species, plasma LPC(16:0) and LPC(18:1) showed strong correlations with sensorimotor dysfunction, lesion severity, and mechanistic biomarkers in the rat stroke model. Despite notable discrepancies between plasma and brain LPC levels, both were strongly linked to functional outcomes and mechanistic biomarkers, suggesting that LPC’s prognostic value is retained extracranially. This study advances the understanding of LPC as a blood marker in ischemic stroke and highlights directions for future research to further elucidate its association with stroke severity, particularly through investigations in more clinically representative models.

## 1. Introduction

Ischemic stroke ranks as the fifth leading cause of morbidity and mortality in the United States [1], yet there is a lack of adequate prognostic biomarkers [2,3,4,5]. Reliable biomarkers are essential for predicting stroke outcomes, guiding treatment decisions, and ultimately improving patient care and long-term prognosis [6,7]. Current clinical standards for stroke detection and subtype differentiation, such as neuroimaging, blood markers, and functional status assessment at intake, are limited by bias and suboptimal specificity and sensitivity [4,5,8,9]. Emerging research points to a growing number of promising blood-based biomarkers for ischemic stroke, including markers of blood–brain barrier (BBB) damage, apoptosis, inflammation, and non-coding RNAs [4,8,10,11,12]. Lipids, in particular, are gaining recognition as critical biomarkers in stroke management due to their fundamental roles in brain structure and function [13,14]. Moreover, lipids are well-established risk factors for ischemic stroke, with age-related imbalances in cholesterol and lipoprotein levels not only increasing stroke incidence but also promoting atherosclerosis, an additional marker of elevated stroke risk [15,16,17]. In ischemic stroke, lipid dysfunction occurs at the onset and acts as an upstream regulator of apoptosis, inflammation, and oxidative stress, all of which are critical contributors to the exacerbation of ischemic brain injury [13,14,18,19,20]. This highlights the potential of lipids as valuable biomarkers in ischemic stroke.

Lipid and phospholipid dysfunction, along with the subsequent release of lipid mediators and bioactive metabolites, such as lysophosphatidylcholine (LPC), is increasingly recognized as a critical contributor to the progression of acute ischemic stroke and as a promising prognostic marker [21,22,23,24,25,26,27]. Recent studies have identified reduced plasma levels of various LPC species during the acute phase as potential indicators of clinical stroke occurrence and poor long-term functional outcomes [6,28]. However, these studies have not yet differentiated the relationship between plasma LPC levels and varying degrees of ischemic severity, leaving the precise strengths and limitations of plasma LPC as a prognostic marker unclear. While plasma LPC has not been extensively studied in preclinical models of ischemic stroke, preclinical research has shown that elevated brain LPC levels are directly associated with lesion development and poorer functional outcomes through increased inflammation, apoptosis, oxidative stress, and BBB damage [23,27,29,30,31,32,33,34,35]. Specifically, LPC has been identified as a conditional danger-associated molecular pattern (DAMP) that triggers sterile inflammation at elevated concentrations [36]. Unlike other blood biomarkers for ischemic stroke—such as those indicating BBB damage, inflammation, and apoptosis [37,38,39,40,41], which typically reflect patterns observed in the brain—a disjunction between blood and brain LPC levels has been observed in brain injuries, including focal and global ischemia, as well as mild and severe traumatic brain injury [23,29,30,42,43,44,45]. To validate plasma LPC as a reliable prognostic marker for stroke and ensure that it reflects disease progression within the brain rather than secondary damage elsewhere in the body, the dynamic relationship between plasma and brain LPC levels needs to be thoroughly investigated.

This study investigated plasma LPC as a potential biomarker across varying severities of ischemic stroke using a rat model of transient middle cerebral artery occlusion (tMCAO), a well-established pre-clinical model despite the anatomical differences in brain structure between rats and humans, particularly the lower white matter-to-gray matter ratio in rats [46,47]. In addition to the sham and tMCAO control groups, we included groups that received LPC supplementation and groups where LPC formation was inhibited by the PLA2 inhibitor. We measured the levels of five main LPC species in plasma, correlating them with structural and functional stroke outcomes, and investigated whether the prognostic value of plasma LPC was specific to any particular LPC species. Concurrently, brain LPC levels were assessed to determine whether plasma LPC accurately reflects changes occurring in the brain following stroke. The prognostic strength of plasma LPC was further validated through comparisons with markers of BBB damage, neutrophil levels, apoptosis, and oxidative stress, all of which have been previously recognized as promising biomarkers for ischemic stroke. Our findings highlight the prognostic potential of two specific LPC species, LPC(16:0) and LPC(18:1), as blood markers in ischemic stroke.

## 2. Results

### 2.1. Decreased Plasma Levels of LPC Exhibit an Inverse Correlation with Brain LPC Concentrations and Lesion Volume in Ischemic Stroke

Levels of five main LPC species (LPC(18:1/17:1), LPC(22:6/17:1), LPC(20:4/17:1), LPC(16:0/17:1), and LPC(18:0/17:1)) were quantified in both plasma and brain tissues using HPLC-MS techniques (Figure 1). Following MCAO induction, plasma concentrations of all five LPC species exhibited a decline in a stepwise manner between 2 and 2.5 h post-induction. Although LPC(22:6/17:1), LPC(20:4/17:1), and LPC(18:0/17:1) levels began to rise 24 h after MCAO, they remained below sham levels, while LPC(18:1/17:1) and LPC(16:0/17:1) continued to decrease. Conversely, all LPC levels in the brain increased significantly (2–3-fold) at 24 h post-MCAO.

A cerebral perfusion assessment revealed a significant reduction in perfusion (~61%) in the ipsilateral hemisphere of the vehicle group at 6 h post-MCAO (Figure 2A,B). These perfusion changes resulted in substantial lesion development at 24 h (total lesion: 25.7 ± 2.5%, core lesion: 10.3 ± 3.5%) (Figure 2C,D). This suggests a direct relationship between brain LPC levels and lesion volume and an inverse relationship between plasma LPC levels and lesion volume in ischemic stroke.

### 2.2. Increased Brain LPC Levels Lead to Impaired Cerebral Blood Flow and a Decrease in Salvageable Tissue Following Ischemic Stroke

To validate the observed direct relationship between brain LPC levels and lesion volumes, as well as the inverse relationship between plasma LPC levels and lesion volumes, one group of rats was administered LPC(18:1), while another received Varespladib, a PLA2 inhibitor. The LPC(18:1) supplementation group showed increased concentrations of LPC(18:1) and LPC(22:6) in the brain (Figure 1 and Figure 3) compared to the vehicle group, while plasma levels remained comparable to those of the vehicle group. In contrast, the administration of Varespladib resulted in decreased concentrations of all LPC species in the brain and increased levels in the plasma.

Cerebral perfusion and lesion analysis further supported the observations made in the vehicle group (Figure 4). LPC(18:1) supplementation led to a further reduction in perfusion to ~48%, while the inhibition of LPC formation with Varespladib significantly preserved cerebral perfusion at ~72%. LPC−treated animals exhibited a total damage area similar to that of the vehicle group (29.2 ± 4.9%) (Figure 2 and Figure 4) but with a significantly larger lesion core (18.3 ± 3.7%). In line with the increased cerebral perfusion, LPC formation inhibition markedly reduced both the total lesion size (17.9 ± 2.1%) and lesion core (6.0 ± 1.2%) at 24 h. These findings confirm the inverse relationship between plasma LPC levels and the overall lesion volume, though not with the lesion core.

### 2.3. Plasma LPC Levels Are Inversely Correlated with Brain LPC Levels Following Ischemic Stroke

To evaluate the relationship between LPC levels in plasma and brain tissue, correlation graphs were generated for each individual LPC species (Figure 5). Overall, plasma LPC and brain LPC demonstrated a negative association. Notably, two LPC species exhibited significantly higher correlation coefficients, with LPC(18:1) showing an R^2^ value of 0.5691 and LPC(16:0) showing an R^2^ value of 0.6075. The correlation coefficients for LPC(22:6), LPC(20:4), and LPC(18:0) were R^2^ = 0.2022, 0.2952, and 0.1145, respectively. 

### 2.4. Both Plasma and Brain LPC Levels Are Strongly Correlated with Sensorimotor Dysfunction Following Ischemic Stroke

Sensorimotor dysfunction was assessed using a combination of the Garcia and proprioception tests. All three tMCAO groups showed increased sensorimotor dysfunction compared to sham-operated controls. The LPC(18:1)-treated group, in particular, exhibited slightly worse performance than the vehicle-treated group and significantly poorer performance than the Varespladib-treated group: Garcia test (sham: 18.0 ± 0.0, vehicle: 8.9 ± 1.2, LPC+: 7.5 ± 0.7, LPC−: 12.2 ± 1.8) (Figure 6A) and Proprioception-contralateral test (sham: 0.0 ± 0.0, vehicle: 8.8 ± 1.8, LPC+: 10.0 ± 1.8, LPC−: 6.5 ± 1.8) (Figure 6B) (*p* < 0.05). 

The relationship between plasma and tissue LPC levels and functional outcomes was evaluated using correlation graphs (Figure 6, Appendix A). Notably, among the LPC species measured in plasma, LPC(18:1) and LPC(16:0) exhibited the strongest correlations. In contrast, tissue LPC analysis revealed that LPC(20:4) had the strongest correlation with functional outcomes, followed closely by LPC(18:1) and LPC(16:0).

### 2.5. Both Plasma and Brain LPC Levels Show a Strong Correlation with Established Mechanistic Biomarkers in Ischemic Stroke

To investigate potential underlying mechanisms, markers of BBB damage (MMP9), neutrophil levels (MPO), apoptosis (cleaved caspase 3/caspase 3 (CC3/C3)), and oxidative stress (Nitrotyrosine) were assessed in penumbral tissue 24 h post-MCAO. MMP9 expression, indicative of BBB damage, significantly increased in all tMCAO groups (sham: 1.0 ± 0.9; vehicle: 7.8 ± 4.7; LPC+: 10.0 ± 4.1; LPC−: 5.2 ± 2.6) (Figure 7A). MPO expression, a marker of neuroinflammation, also rose in the vehicle and LPC+ groups (sham: 1.0 ± 0.3; vehicle: 4.1 ± 2.1; LPC+: 10.4 ± 4.9; LPC−: 1.4 ± 0.8) (Figure 7B). The CC3/C3 ratio, indicative of apoptosis, increased in all tMCAO groups (sham: 1.0 ± 0.1; vehicle: 3.1 ± 1.0; LPC+: 3.0 ± 1.0; LPC−: 2.1 ± 0.4) (Figure 7C). Nitrotyrosine expression, reflecting oxidative stress, significantly increased only in the vehicle and LPC groups (sham: 1.0 ± 0.6; vehicle: 2.1 ± 0.6; LPC+: 2.0 ± 1.0; LPC−: 1.2 ± 0.1) (Figure 7D). LPC inhibition led to a significant reduction in the expression of all four markers.

The relationship between all four biomarkers and plasma and brain LPC levels was evaluated using correlation graphs (Figure 7, Appendix A. MMP9 and MPO showed stronger correlations with plasma LPC(16:0) and LPC(18:1) (MMP9: R^2^ = 0.5079, 0.3575; MPO: R^2^ = 0.5425, 0.4029) compared to the other two markers. The CC3/C3 ratio demonstrated a moderate correlation with LPC(16:0) and LPC(18:1), while Nitrotyrosine exhibited significantly lower correlations (Appendix A). All other LPC species displayed weaker correlations with all the biomarkers.

## 3. Discussion

This study demonstrates the potential of LPC(16:0) and LPC(18:1) as prognostic blood biomarkers across a range of ischemic severities, from a mild ischemic burden in animals with LPC inhibition to severe ischemic burden in animals supplemented with LPC(18:1). While all five LPC species decrease in plasma and increase in the brain, LPC(16:0) and LPC(18:1) exhibit the strongest correlations with both functional and structural stroke outcomes. These LPC species also demonstrate strong associations with brain markers of BBB damage, neutrophil levels, apoptosis, and oxidative stress, all of which are established prognostic indicators. Despite the discordance between plasma LPC levels (which decrease) and brain LPC levels (which increase), both are strongly correlated with functional outcomes and supported by underlying mechanisms, highlighting the association between plasma LPC and the pathological development of ischemic stroke. Overall, our findings underscore the potential of plasma LPC(16:0) and LPC(18:1) as prognostic markers in ischemic stroke, demonstrating strong correlations with both functional and structural outcomes, as well as with previously validated blood markers.

Our results identify plasma LPC(16:0) and LPC(18:1) as promising prognostic markers in ischemic stroke. These markers reflect varying levels of ischemic damage: sham-operated animals show no damage, vehicle animals exhibit a moderate stroke burden, LPC−inhibited animals show a mild burden, and LPC−supplemented animals present a severe burden. Although all five plasma LPC species decreased following MCAO, LPC(16:0) and LPC(18:1) demonstrated the strongest correlations with sensorimotor functions and are highly associated with lesion volumes. These are also the only LPC species for which plasma levels continue to decrease between 2 and 24 h after MCAO. Furthermore, comparisons with established prognostic markers show that LPC(16:0) and LPC(18:1) correlate significantly with markers of BBB damage, neutrophil infiltration, apoptosis, and oxidative stress [37,38,39,40,41]. Clinically, decreased plasma LPC(16:0) is consistently observed in the acute phase of ischemic stroke compared to normal controls [48,49], and low levels of LPC(16:0) and LPC(18:1) within the first 24 h are indicative of poor recovery after 3 months [6]. Although plasma LPC(20:4) has been proposed as a potential long-term outcome predictor [50], it did not show the same strength or consistency as LPC(16:0) and LPC(18:1) in our study. This discrepancy may result from inter-species variation or from the fact that our assessments were conducted at 24 h, whereas clinical evaluations are performed at 3 months. Although LPC(16:0) and LPC(18:1) exhibit strong specificity for ischemic stroke, our findings also reveal limitations in their sensitivity for distinguishing stroke severity. Plasma LPC levels did not differ significantly between the vehicle (moderate) and LPC supplementation (severe) groups. However, there was more than an 80% difference in the core (unsalvageable) lesion tissue and approximately a 15% decrease in sensorimotor function in the LPC supplementation group. Plasma LPC is effective in differentiating between sham, mild, and moderate-to-severe stroke collectively but is less effective in distinguishing between moderate and severe stroke severities. Further research is needed to fully understand the extent to which plasma LPC(16:0) and LPC(18:1) reflect the stroke burden. 

We have demonstrated a notable disjunction between plasma and brain LPC levels; however, both show strong correlations with functional outcomes and established mechanistic biomarkers. Notably, plasma LPC correlates with mechanistic biomarkers with a strength comparable to brain LPC, suggesting that the prognostic value is preserved when using an extracranial marker. LPC(16:0) and LPC(18:1) emerge as particularly promising markers, exhibiting the highest plasma–brain correlations. Clinically, blood markers are routinely employed to monitor disease progression [4,8,9], with plasma levels of markers for BBB damage, apoptosis, and neutrophil presence generally reflecting trends observed in the brain [37,38,39,40,41]. However, this correlation is inconsistent with previous findings of LPC in focal and mild global ischemia, as well as mild and severe traumatic brain injury [23,29,30,42,43,44,45], highlighting the need for the simultaneous measurement of plasma and brain LPC levels. Currently, simultaneous assessments of plasma and brain LPC levels are limited to preclinical studies using cardiac arrest [51] and mild global ischemia models [45]. In cardiac arrest models, LPC(22:6) levels decreased in both plasma and brain at 2 and 4 h post-event [51], although a prior study reported an increase in brain LPC levels [52]. Conversely, in mild global ischemia, several LPC species decreased in the brain while increasing in the plasma [45]. These inconsistencies, even within similar models, suggest that LPC patterns are context-specific, influenced by the disease type and severity, which may explain the deviations observed in our results. In ischemic stroke, previous preclinical research has shown increased LPC levels in the brain associated with greater neurological deficits and reduced motor function [23,32]. The absence of simultaneous plasma and brain LPC measurements in the literature has left a gap in understanding why plasma levels may decrease while brain levels increase concurrently.

Plasma LPC levels are influenced by several factors, including the rates of de novo synthesis and hydrolytic degradation [34,53,54]. Specifically, LPC hydrolysis in plasma is mediated by enzymes such as lysophospholipase A1, which is present in neutrophils [34]. Clinically, poor outcomes in ischemic stroke correlate with increased plasma neutrophil levels [55], leading to reduced plasma LPC due to enhanced hydrolysis. However, the synthesis and hydrolysis rates alone do not account for the increased LPC in the brain and decreased levels in plasma. It is plausible that, during brain injury, the rate of LPC uptake into the brain exceeds its rate of recirculation, thereby reducing plasma LPC availability. Patterns of elevated brain LPC and decreased plasma LPC, similar to those observed in mild traumatic brain injury [42,43], indicate that LPC may not only function as a DAMP [36,56,57], but also be drawn to other DAMPs released during brain injury. Furthermore, the breakdown of the Lands cycle in the ischemic brain, which regulates the conversion between phosphatidylcholine (PC) and LPC, could lead to increased PC-to-LPC conversion, decreased reverse conversion, and a consequent reduction in the plasma LPC concentration [22,58,59]. This may explain why LPC(18:1) supplementation does not yield the same benefits as its precursor fatty acid, oleic acid, in ischemic conditions [60,61,62,63,64,65]. Further research is needed to elucidate the predominant factors in ischemic stroke. 

Plasma LPC(16:0) and LPC(18:1) demonstrate strong correlations with various molecules previously validated as prognostic markers in ischemic stroke, including those related to neutrophil infiltration and BBB disruption. Although plasma levels of these markers were not directly measured, they typically correlate closely with plasma LPC levels [37,38,39,40,41]. LPC is known to enhance inflammatory and atherogenic processes [27,33,34,35,36,56,57], which are associated with improved prognostic value in ischemic stroke [4]. Specifically, LPC(16:0) is linked to inflammation, T-cell activation, and increased expression in conditions, such as obesity and diabetes [22,33,66,67], while both LPC(18:1) and LPC(16:0) are associated with elevated cytokine levels [33]. Additionally, LPC(16:0) and LPC(18:1) correlate strongly with brain MMP-9 levels. Plasma MMP-9 increases due to its release from the brain into the bloodstream, and elevated MMP-9 levels have been associated with improved prognostic value in clinical studies [4,68]. This is related to increased BBB breakdown mediated by LPC−induced interleukin-1β release, which enhances MMP-9 production from oligodendrocytes [69]. Furthermore, LPC(16:0) and LPC(18:1) are strongly correlated with the CC3/C3 ratio, indicating a connection to apoptosis development. While the caspase-3 content in blood has been evaluated as a diagnostic marker for ischemic stroke, its prognostic value remains limited [4,70]. Nonetheless, apoptosis is a major factor in ischemic stroke pathogenesis [71], and LPC contributes to apoptosis progression through caspases, cytochrome C, calcium influx, and the mitochondrial pathway [33,72,73]. Lastly, plasma LPC(16:0) and LPC(18:1) are associated with prostacyclin production, which has beneficial effects in ischemic stroke [48,49,74,75]. Decreased plasma levels of these LPC species reduce prostacyclin production, exacerbating ischemic damage by promoting platelet microthrombi, BBB damage, and vasoconstrictor release. Conversely, lipid dysregulation and membrane breakdown in the brain lead to inhibition of in situ prostacyclin synthesis [48]. Lipid dysregulation, neuroinflammation, and myelin breakdown are pivotal contributors to stroke pathogenesis, leading to cognitive decline and neurobehavioral dysfunction [13]. Therefore, the development of blood tests focusing on specific lipids, particularly LPC(16:0) and LPC(18:1), is crucial for enhancing prognostic accuracy and for evaluating clinical trial outcomes.

This study has several limitations. Firstly, additional time points for plasma LPC measurements could offer deeper insights into the relationship between plasma LPC levels and stroke severity. Plasma LPC was assessed only at 2, 2.5, and 24 h post-MCAO, with lesion analysis conducted concurrently with the final plasma measurements used for correlational assessments. The observed stepwise decrease in LPC levels suggests that there may be an optimal time point for plasma LPC measurements that better correlates with the stroke burden and possesses prognostic value. Furthermore, assessing LPC levels at a later time point, such as 72 h post-stroke, could provide additional insights into the relationship between LPC and the prolonged degradative mechanisms associated with ischemic stroke, including inflammation and edema [76,77,78]. Secondly, stroke severity was assessed exclusively at the 24 h mark; longer-term evaluations, extending over weeks to months, may more accurately reflect chronic outcomes, as clinical studies typically emphasize prolonged timeframes, often spanning several months [6,79,80]. Thirdly, brain LPC levels were measured only in the ipsilateral penumbra, the most salvageable region of the ischemic brain [81]. Measurements from the lesion core or contralateral hemisphere could provide further insight into the relationship between brain LPC and lesion development, as well as the interaction between plasma LPC and lesion progression. Lastly, while the tMCAO model is a well-established rodent model of ischemic stroke [46,47], it does not fully mirror clinical conditions, particularly due to reduced white matter injury compared to humans [82] and the use of younger rats, despite age-related dyslipidemia being a key independent risk factor for ischemic stroke [15,16,17].

In conclusion, our study highlights the potential of plasma LPC(16:0) and LPC(18:1) levels as prognostic markers in ischemic stroke, offering the first exploration of the brain-blood LPC disjunction to elucidate underlying mechanisms. The strong correlation between plasma LPC levels and structural and functional outcomes in a rodent model of ischemic stroke, as well as their interaction with downstream damage mediators, emphasizes the need for further investigation into the interplay between brain and plasma LPC. While these markers demonstrate high specificity, their sensitivity remains uncertain, necessitating additional assessments to evaluate their effectiveness in distinguishing between moderate and severe ischemic stroke outcomes. Furthermore, interspecies variation and age-related risk factors emphasize the importance of examining the relationship between plasma LPC levels and stroke severity in more clinically relevant models to confirm these findings. Overall, this study advances the understanding of LPC’s potential as a prognostic blood marker and outlines future research directions to refine its relationship with ischemic stroke burden.

## 4. Materials and Methods

### 4.1. Animals

Experiments within this study were approved by the Institutional Animal Care and Use Committee of the Feinstein Institutes for Medical Research and performed in accordance with the National Institutes of Health Guidelines For The Use Of Experimental Animals and ARRIVE guidelines. Male Sprague–Dawley rats weighing 350–450 g were used (Charles River Laboratories, Fairfield, NJ, USA). Animals were housed in a temperature-controlled room (12 h light/dark cycle), in cages lined with Enrich-o’Cobbs bedding (The Anderson, Inc., Maumee, OH, USA) and were given access to food and water ad libitum.

### 4.2. Experimental Groups and Exclusion Criteria

A total of 69 age-matched animals were used in this experiment (Sprague–Dawley, 9–10 weeks old, Charles River Laboratories, NY), split across 4 groups (sham, vehicle, LPC, PLA2i). Group sample sizes were planned based on a power calculation (α = 0.05 and β = 0.8) generated using previous results, indicating a minimum group size of n = 6. Sample sizes for neurobehavioral assessments were increased intentionally (n = 10) to account for a higher degree of inter-animal variability, based on prior experimental observations. Animals undergoing laser speckle were used for lesion analysis via triphenyltetrazolium chloride (TTC) assessment, and separate groups were used for Western blotting. A total of 7 animals were excluded from all analyses; 2 animals experienced violation of the dura mater during laser speckle assessment, 3 animals died prior to 3 h, and 2 sham animals were excluded due to failing the rotarod assessment. A total of 14 animals from the remaining 62 died prior to 24 h.

### 4.3. Transient Middle Cerebral Artery Occlusion

Animals underwent tMCAO induction with 120 min occlusion (Figure 8A). Rats were anesthetized via isoflurane in air and animal body temperatures were maintained at 37 ± 1 °C using a heating blanket. A cervical incision was achieved on the ventral side, keeping the incision laterally toward the right region. The underlying tissues were carefully dissected to locate the right common carotid artery (CCA). The external branch of the CCA was identified, released from the surrounding tissues, ligated with a permanent knot with black silk (6/0) above the origin of superior thyroid artery, and cut using a sharp scissor near the bifurcating point. A nylon suture (3/0) with a silicone tip and a length of 3 cm (403534, Doccol Corporation, Sharan, MA, USA) was inserted into the external carotid artery and advanced further into the internal carotid artery to the origin of middle cerebral artery (MCA), where resistance indicated occlusion of the MCA. The suture was loosely fixed in place with black silk. Incisions were closed with staples and the rats were returned to their home cages. After 120 min, rats were re-anesthetized, the filaments were removed, and incisions were closed with non-absorbable nylon sutures. Animals had 300 µL of blood drawn from an intravenous catheter at baseline, reperfusion, and then at 30 min post reperfusion. The sham group was exposed to a similar surgery but without cutting of the external carotid artery and suture insertion. All rats received buprenorphine and were returned to their home cages for 24 h observation.

### 4.4. Drug Administration

LPC(18:1) (Cayman Chemical, Ann Arbor, MI, USA) and Varespladib (Selleck Chem, Houston, TX, USA) were administered via intravenous and subcutaneous injections, respectively, after a 30 min blood draw. Animals were administered 6 mg/kg LPC(18:1) in 0.5% bovine serum albumin (BSA) in 1X phosphate-buffered saline (PBS) (Sigma, Burlington, MA, USA). Vehicle animals were administered a solution of 0.5% BSA in 1X PBS. Varespladib was administered as a dosage of 8 mg/kg in a solution (5% DMSO, 5% Tween-20, 20% PEG, 70% normal saline). 

### 4.5. Laser Speckle Flowmetry

Laser speckle perfusion imaging of the brain was performed using an RFLSI III Laser Speckle Imaging System according to the manufacturer’s instructions. Briefly, a midline scalp incision was made to an isoflurane-anaesthetized rat and the skull was exposed; the camera of LSF was installed 30 cm above the skull using an articulating arm. Two identical rectangular regions of interest were selected on each of the 2 hemispheres. The imaging was set up at a display rate of 25 Hz, time constant of 1 s, and a camera exposure time of 3 milliseconds. A 2 min baseline CBF flux was recorded before the induction of ischemia. MCAO surgery was performed and CBF was measured at 6 h post-ischemia. Data were expressed as mean CBF flux at 6 h post ischemia. 

### 4.6. Neurobehavioral Assessments

Neurobehavioral assessments occurred 24 h following MCAO or sham surgery. Assessments followed previously established protocols and were performed in order of least stressful to most stressful. Tests were performed in a dedicated assessment suite, with consistent lighting and minimal extraneous stimuli. Assessments were performed by a dedicated technician blinded to the experimental groups. Testing surfaces were cleaned with 75% ethanol solution before and after each trial to prevent scent trails.

#### 4.6.1. Garcia Score

General sensorimotor function was assessed using the Garcia score, which is specifically designed to assess unilateral injuries in MCAO. The Garcia test consists of a 6-factor scale with each factor having a maximum score of 3, and a total maximum score of 18. The components measured are spontaneous activity, symmetry in limb movement, forepaw outstretching, climbing, body proprioception, and response to vibrissae touch.

#### 4.6.2. Proprioception

Proprioception was assessed via an analysis of the rats’ reaction to visual and tactile stimuli. The visual proprioception assessment involved moving the rat in visual range of a beam and seeing whether it would reach for the beam. Tactile perception involved placing either the hindlimbs or forelimbs atop the beam and seeing whether the rat would react to the tactile sensation. Contralateral and ipsilateral proprioception were assessed separately, for a total score of 12 on each side.

### 4.7. Sample Collection and Processing

At the time of final sample collection, animals were heavily anesthetized using isoflurane, had blood collected from the vena cava, and were then decapitated. Blood was stored at 4 °C for 30 min and then spun at 2000 G for 10 min. The plasma was then collected and frozen at −20 °C until further usage. The brains from animals that underwent laser speckle were used for lesion analysis, while another set of animals (n = 6/group) was used for biochemical analyses. Samples for biochemical analyses included the ipsilateral hemisphere up 0–4 mm from the midline, in order to exclude the core (Figure 8B). Samples were flash frozen and powdered in liquid nitrogen and then stored until usage at −80 °C. 

### 4.8. TTC Staining and Lesion Assessment

To measure the lesion volume, brains were serially sectioned at a distance of 2 mm and stained with 2.0% triphenyltetrazolium chloride (TTC) (Sigma, Burlington, MA, USA) for a period of 30 min at 37 °C. The sections were then removed from the TTC, washed, and placed in 10% formalin (Sigma, Burlington, MA, USA) overnight for preservation. Digital images of the brain slices were acquired using the Olympus SZ2-STS stereomicroscope equipped with an Olympus DP27 camera (Olympus Life Science, Waltham, MA, USA). The lesion volume was calculated manually using the ImageJ (Version 1.54) tracing tool for each section. The penumbra and core were quantified separately, with the core defined as tissue that had turned white upon TTC staining. Section areas were summed, and the %lesion was calculated using the area of the whole brain slice.

### 4.9. LPC Analysis via HPLC-MS

LPCs were extracted from brain and plasma samples using an established method [51,63]. Briefly, pulverized brain tissue (9 mg) was extracted using 50 μL of potassium phosphate buffer (100 mM, pH 7.4) in combination with 950 μL of CHCl_3_:MeOH (2:1, v:v) containing butylated hydroxytoluene (2 mM). Frozen plasma (50 μL) was extracted using 750 μL of ethanol. The extracts were evaporated and reconstituted in 200 μL of isopropanol:t-butyl methyl ether:aqueous ammonium formate, at 34:17:5 v:v:v. The samples were then analyzed using mass spectrometry. Data processing was completed using Xcalibur software (version 2.2; Thermo Scientific, Waltham, MA, USA).

### 4.10. Western Blotting

Powdered tissue samples were aliquoted and homogenized in RIPA lysis buffer (Thermo Fisher Scientific, Waltham, MA, USA) containing a protease and phosphatase inhibitor cocktail (Thermo Fisher Scientific, Waltham, MA, USA) with a bead mill homogenizer. The protein concentration was quantified using the BCA protein assay kit (Thermo Fisher Scientific, Waltham, MA, USA). Protein samples were separated on a 4–20% TGX gel (Bio-Rad, Hercules, CA, USA) via electrophoresis, according to molecular weight, using the Protein Plus Kaleidoscope (Bio-Rad, Hercules, CA, USA) for visualization. Proteins were then electro-transferred onto polyvinylidene difluoride membranes using the semi-dry transfer method, blocked with 5% skimmed milk (Thermo Fisher Scientific, Waltham, MA, USA) in 1X Tris-buffered saline with Tween 20 (TBST) (Millipore Sigma, Burlington, MA, USA) at room temperature for 1 h, and then incubated with primary antibodies (anti-caspase-3/cleaved-caspase-3 (C3-CC3) (ms, 1:1000, Proteintech, Rosemont, IL, USA), anti-Nitrotyrosine (NT) (ms, 1:1000, Abcam, Waltham, MA, USA), anti-myeloperoxidase (MPO) (rb, 1:1000, 1:1000, Abcam, Waltham, MA, USA), anti-matrix metalloproteinase-9 (MMP-9) (rb, 1:1000, Abcam, Waltham, MA, USA)) overnight at 4 °C. After three washes in TBST, the membranes were incubated with respective HRP-conjugated secondary antibodies (Abcam, Waltham, MA, USA) in 5% milk at room temperature for 1 h, followed by another three washes with TBST. Signals were detected via chemiluminescence using ECL substrate (Thermo Fisher Scientific, Waltham, MA, USA) on a Bio-Rad ChemiDoc Imaging System. ImageJ (Version 1.54) was then used to quantify relative protein levels in the blots. Any changes to brightness or contrast for the purpose of visualization were consistent across entire blots. β-actin (ms, 1:25,000, Sigma, Burlington, MA, USA) was used as a loading control. All Western blot data are expressed as a ratio of the sham.

### 4.11. Statistical Analyses

Data are represented as the mean ± standard deviation (SD), with individual points present on bar graphs. HPLC-MS time-lapse data are presented as means ± standard errors of the mean (SEM) on a scatter plot. Normal distributions were verified using the Shapiro–Wilk and the Kolmogorov–Smirnov tests. Significant differences were assessed using a one-way analysis of variance test (ANOVA) followed by Fisher’s least significant difference test. HPLC-MS time-lapse data were assessed using one-way repeated measures ANOVA, followed by the Bonferroni test. Data were considered significantly different at *p* < 0.05. All statistical analyses were performed using GraphPad Prism (version 9, GraphPad Software, Boston, MA, USA).

## Figures and Tables

**Figure 1 ijms-25-10649-f001:**
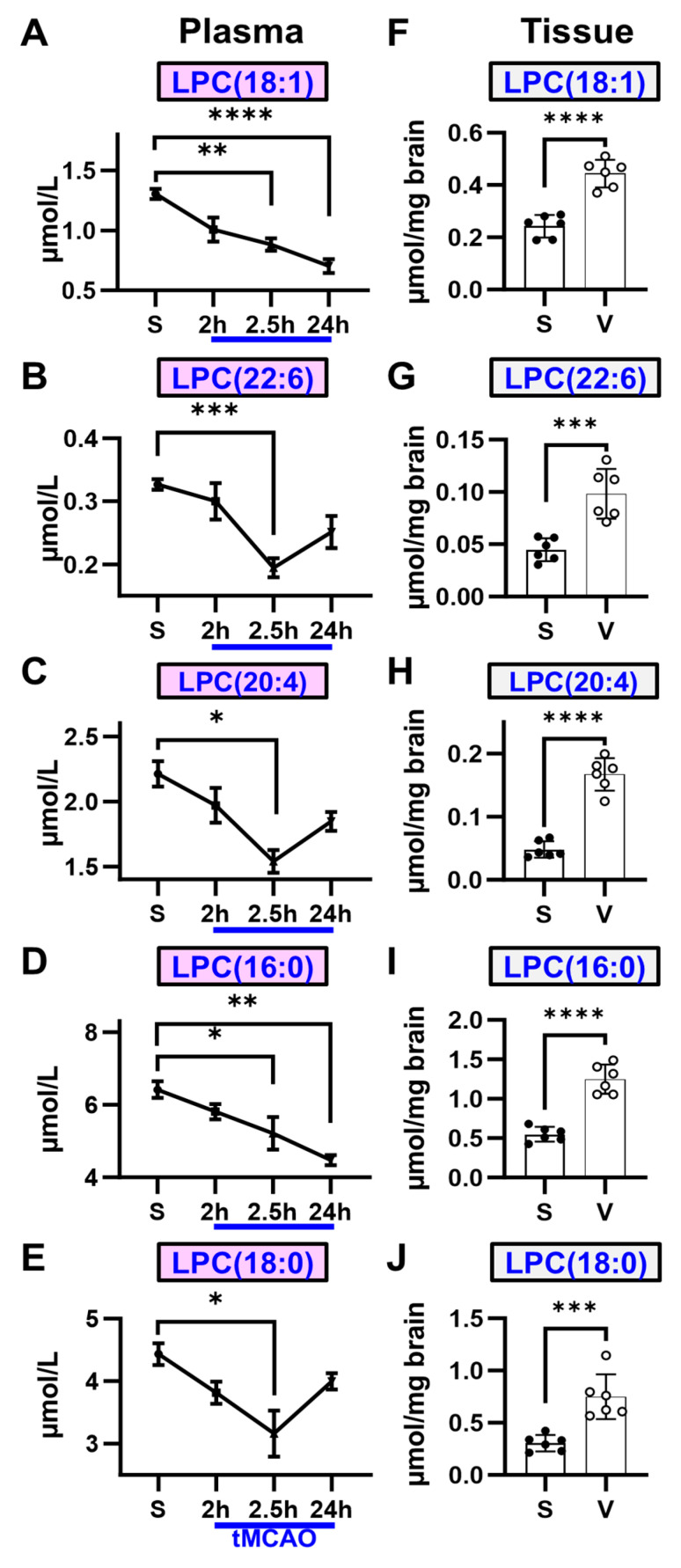
Plasma LPC levels decrease, while brain LPC levels increase following ischemic stroke. (**A**–**E**) Plasma concentrations of all five LPC species progressively decline following MCAO and reperfusion. Notably, LPC(18:1) and LPC(16:0) show a marked decrease between 2 and 24 h, with significant differences from sham levels observed at the 24 h mark. (**F**–**J**) Conversely, all five LPC species exhibit a significant increase in brain tissue at 24 h post-MCAO. (n = 6, * *p* < 0.05, ** *p* < 0.01, *** *p* < 0.001, **** *p* < 0.0001).

**Figure 2 ijms-25-10649-f002:**
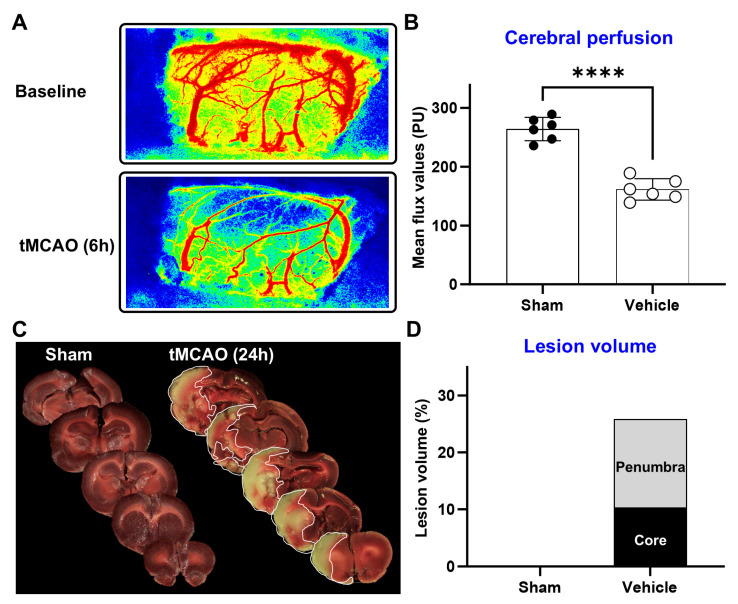
Ischemic stroke leads to reduced cerebral perfusion and induces substantial lesion formation during the acute phase. (**A**,**B**) At 6 h post-MCAO induction, cerebral perfusion is significantly diminished. (**C**,**D**) By 24 h post-MCAO, a significant lesion develops on the ipsilateral side, encompassing a large portion of core tissue (n = 6, **** *p* < 0.0001).

**Figure 3 ijms-25-10649-f003:**
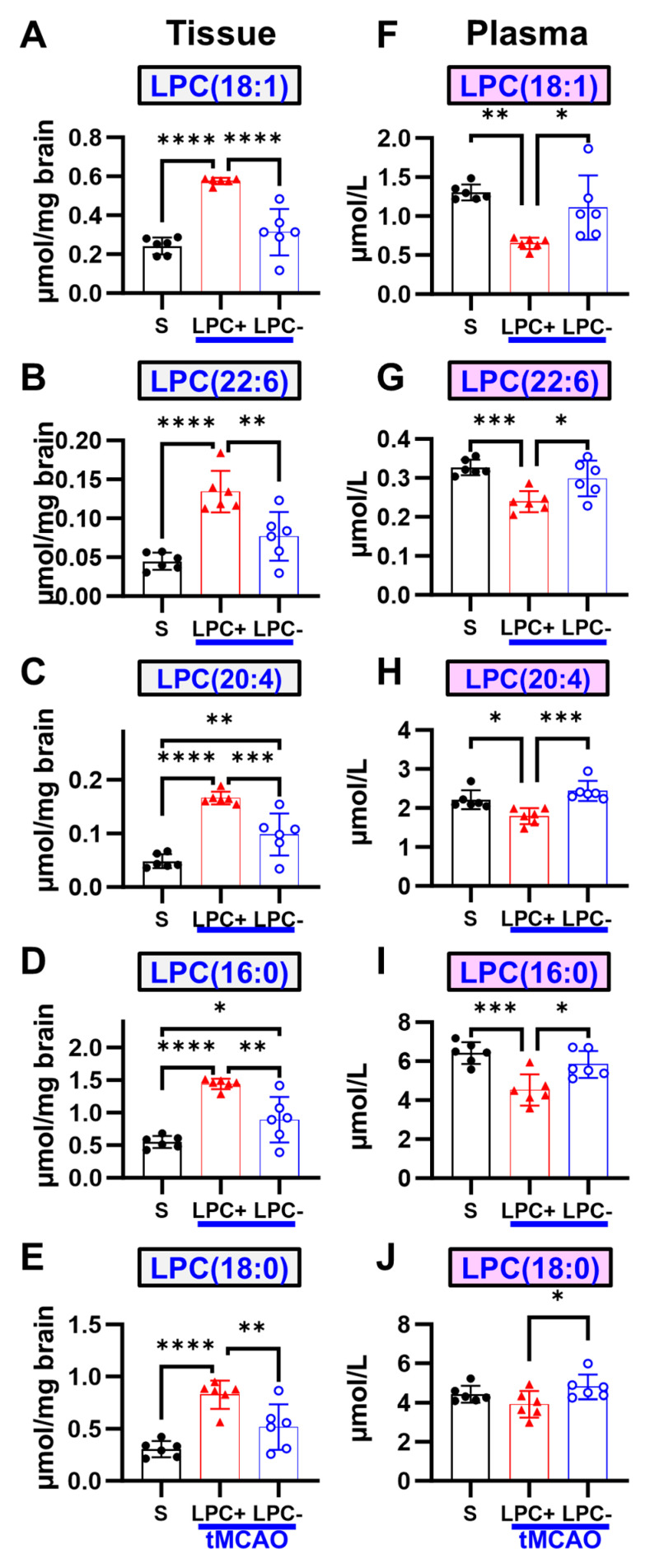
LPC supplementation and inhibition of LPC formation influence both plasma and brain LPC levels following ischemic stroke. (**A**–**E**) Brain levels of all LPC species are elevated above vehicle levels with LPC(18:1) supplementation at 24 h, whereas the inhibition of LPC formation leads to a reduction in all species. (**F**–**J**) In plasma, LPC levels do not fall below vehicle levels at 24 h with supplementation, while inhibition results in increased plasma LPC levels. (n = 6, * *p* < 0.05, ** *p* < 0.01, *** *p* < 0.001, **** *p* < 0.0001, black circles = sham, red triangles = LPC+, blue open circles = LPC−).

**Figure 4 ijms-25-10649-f004:**
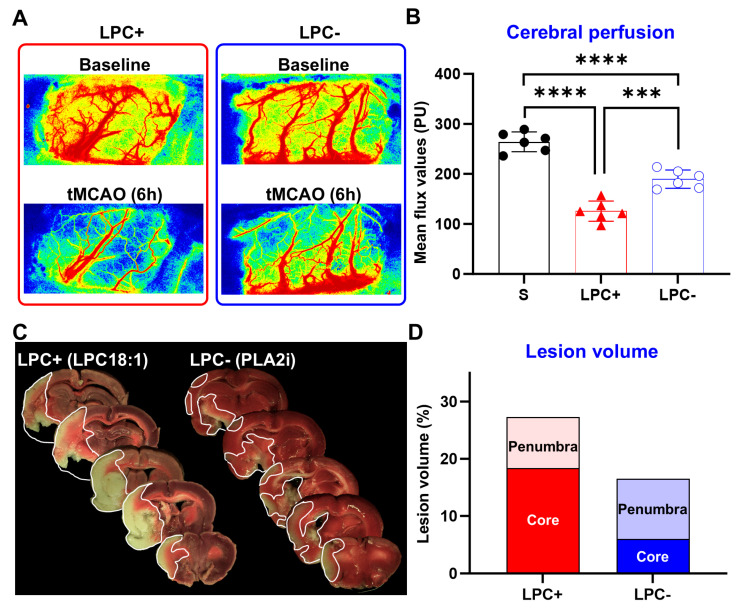
LPC supplementation and inhibition of LPC formation modulate cerebral perfusion and lesion volume following ischemic stroke. (**A**,**B**) LPC supplementation exacerbates the reduction in cerebral perfusion at 6 h, while the inhibition of LPC formation mitigates the perfusion decrease. (**C**,**D**) At 24 h post-MCAO, LPC supplementation results in an increase in the core lesion volume without altering the total lesion volume compared to the vehicle group, while the inhibition of LPC formation reduces both total and core lesion volumes. (n = 6, *** *p* < 0.001, **** *p* < 0.0001, black circles = sham, red triangles = LPC+, blue open circles = LPC−).

**Figure 5 ijms-25-10649-f005:**
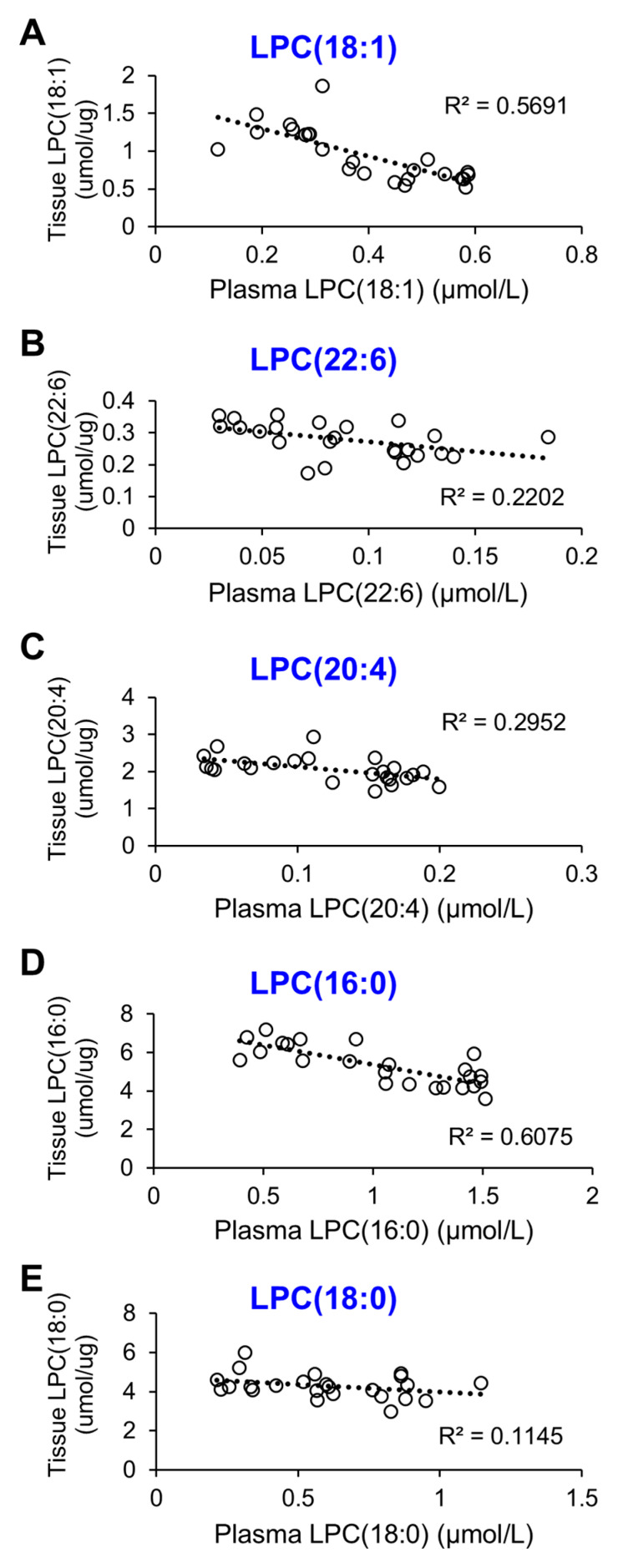
Plasma LPC levels display a negative correlation with brain LPC levels across all five species, with an increase in brain LPC corresponding to a decrease in plasma LPC. Notably, LPC(18:1) and LPC(16:0) are the only species that exhibit a strong correlation across all LPC species. (**A**) LPC(18:1). (**B**) LPC(22:6). (**C**) LPC(20:4). (**D**) LPC(16:0). (**E**) LPC(18:0).

**Figure 6 ijms-25-10649-f006:**
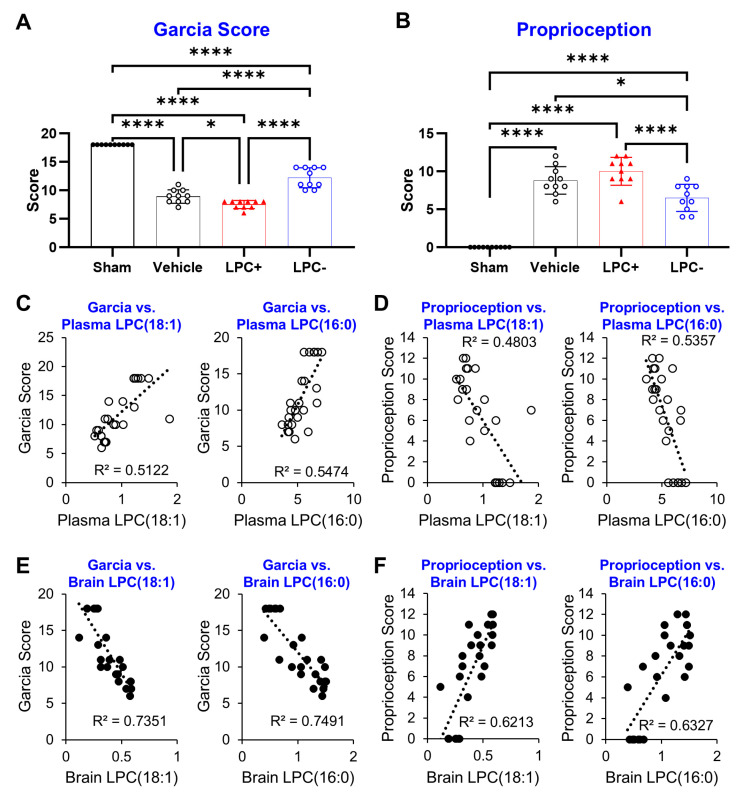
Plasma LPC levels are strongly associated with sensorimotor dysfunction. (**A**,**B**) At 24 h post-MCAO induction, rats show significant sensorimotor and proprioceptive impairments, which are exacerbated by LPC(18:1) supplementation and mitigated by LPC formation inhibition. (**C**,**D**) Plasma LPC(18:1) and LPC(16:0) levels demonstrate moderate correlations with sensorimotor outcomes. (**E**,**F**) Brain LPC(18:1) and LPC(16:0) levels show strong correlations with sensorimotor outcomes. (n = 6, * *p* < 0.05, **** *p* < 0.0001, Bar graphs: black circles = sham, red triangles = LPC+, blue open circles = LPC−, Correlation graphs: black circles = Plasma LPC, open circles = Brain LPC).

**Figure 7 ijms-25-10649-f007:**
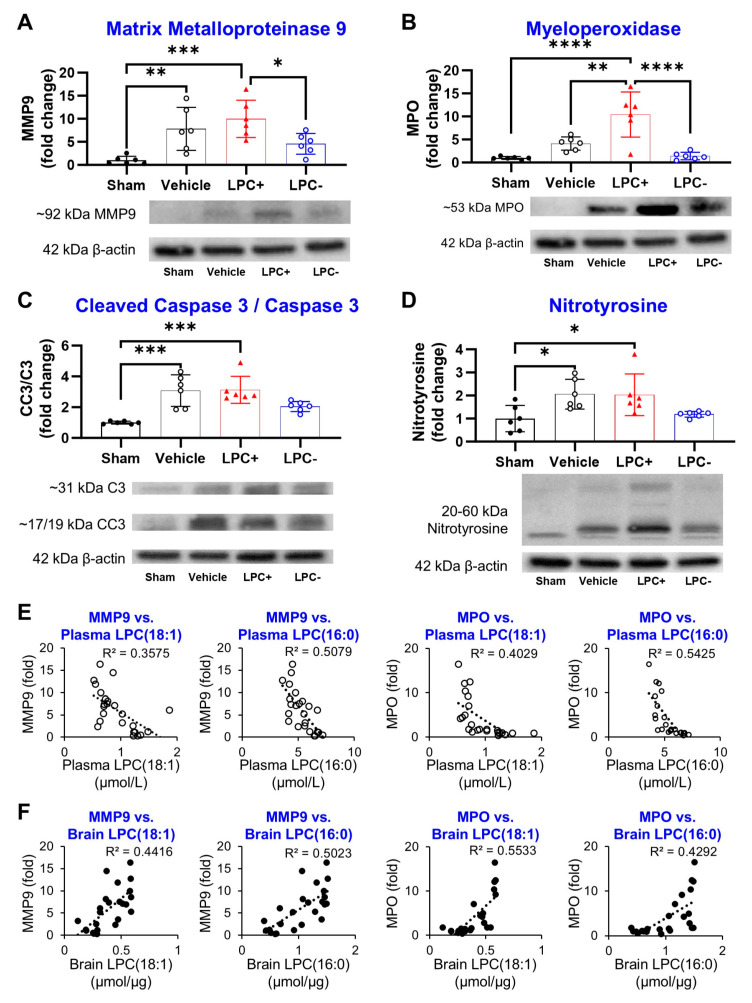
Plasma LPC levels in ischemic stroke are correlated with established prognostic markers. (**A**–**D**) At 24 h post-MCAO induction, brain levels of MMP-9, MPO, CC3/C3, and Nitrotyrosine significantly increase, reflecting BBB damage, neutrophil infiltration, apoptosis, and oxidative stress, respectively. (**E**,**F**) Both plasma and brain levels of LPC(18:1) and LPC(16:0) display moderate correlations with MMP-9 and MPO expression in the brain. (n = 6, * *p* < 0.05, ** *p* < 0.01, *** *p* < 0.001, **** *p* < 0.0001, Bar graphs: black circles = sham, red triangles = LPC+, blue open circles = LPC−, Correlation graphs: black circles = Plasma LPC, open circles = Brain LPC).

**Figure 8 ijms-25-10649-f008:**
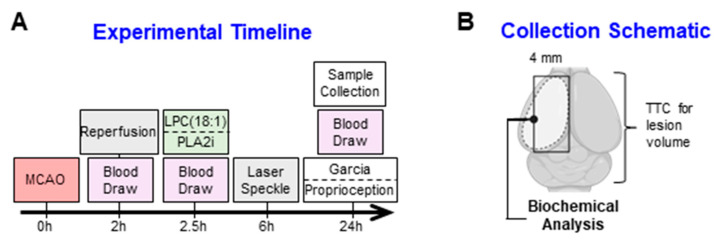
Conceptual diagram of the experimental protocol. (**A**) The timeline illustrates the MCAO and drug-administration procedures. Animals were subjected to 2 h of ischemia followed by reperfusion. At 2.5 h post-MCAO induction, animals received either LPC(18:1) or a PLA2 inhibitor. Cerebral perfusion was assessed 6 h after ischemia, and sensorimotor function was evaluated immediately before sample collection at 24 h post-MCAO. (**B**) Samples for biochemical analyses were collected from the ipsilateral penumbra, with lesion analysis performed using TTC staining.

## Data Availability

The raw data supporting the conclusions of this article will be made available by the authors on request.

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
