# Peer review of "Exploring Lysophosphatidylcholine as a Biomarker in Ischemic Stroke: The Plasma–Brain Disjunction"

_ijms, 2024, doi:10.3390/ijms251910649_

Round 1
Reviewer 1 Report
Comments and Suggestions for Authors
This study investigated the potential of plasma lysophosphatidylcholine (LPC) as a biomarker for various severities of ischemic stroke, using a validated rat model of transient middle cerebral artery occlusion (tMCAO). Alongside the sham and tMCAO control groups, additional groups were included: one receiving LPC supplementation and another where LPC production was inhibited by a PLA2 inhibitor. We quantified five major LPC species in plasma, correlating these levels with both structural and functional outcomes of stroke. We also assessed whether the predictive value of plasma LPC was linked to specific LPC species. Simultaneously, brain LPC levels were analyzed to evaluate whether plasma LPC levels accurately reflected changes in the brain post-stroke. The prognostic capacity of plasma LPC was further validated by comparing it with markers of blood-brain barrier damage, neutrophil levels, apoptosis, and oxidative stress, all of which are recognized as potential biomarkers for ischemic stroke. Our results underscored the prognostic potential of two specific LPC species—LPC(16:0) and LPC(18:1)—as blood-based markers for ischemic stroke. Unfortunately, the authors have overlooked, the single most important risk factor for stroke, the advanced age and associated comorbidities; they are advised to search for biomarkers related to this topic
Comments on the Quality of English LanguageEnglish is, thanks to ChatGPT, OK.
Author Response
Reviewer #1:
This study investigated the potential of plasma lysophosphatidylcholine (LPC) as a biomarker for various severities of ischemic stroke, using a validated rat model of transient middle cerebral artery occlusion (tMCAO). Alongside the sham and tMCAO control groups, additional groups were included: one receiving LPC supplementation and another where LPC production was inhibited by a PLA2 inhibitor. We quantified five major LPC species in plasma, correlating these levels with both structural and functional outcomes of stroke. We also assessed whether the predictive value of plasma LPC was linked to specific LPC species. Simultaneously, brain LPC levels were analyzed to evaluate whether plasma LPC levels accurately reflected changes in the brain post-stroke. The prognostic capacity of plasma LPC was further validated by comparing it with markers of blood-brain barrier damage, neutrophil levels, apoptosis, and oxidative stress, all of which are recognized as potential biomarkers for ischemic stroke. Our results underscored the prognostic potential of two specific LPC species—LPC(16:0) and LPC(18:1)—as blood-based markers for ischemic stroke. Unfortunately, the authors have overlooked, the single most important risk factor for stroke, the advanced age and associated comorbidities; they are advised to search for biomarkers related to this topic.
- We appreciate the reviewer’s thorough evaluation and feedback on our manuscript. We acknowledge that age and its comorbidities were an independent risk factor for stroke development and have accordingly included a series of statements to address this issue. Given that the primary focus of this manuscript is on LPC as a prognostic marker in stroke development, we have contextualized these statements in relation to the observed increase in dyslipidemia with age and its contributory role in stroke incidence. All changes have been highlighted in blue in the main text.
- Lines 46-49: “Moreover, lipids are well-established risk factors for ischemic stroke, with age-related imbalances in cholesterol and lipoprotein levels not only increasing stroke incidence but also promoting atherosclerosis, an additional marker of elevated stroke risk [15–17].”
- Lines 348-351: “Lastly, while the tMCAO model is a well-established rodent model of ischemic stroke [46,47], it does not fully mirror clinical conditions, particularly due to reduced white matter injury compared to humans [82] and the use of younger rats, despite age-related dyslipidemia being a key independent risk factor for ischemic stroke [15–17].”
- Lines 360-363: “Furthermore, interspecies variation and age-related risk factors emphasize the importance of examining the relationship between plasma LPC levels and stroke severity in more clinically relevant models to confirm these findings."

Reviewer 2 Report
Comments and Suggestions for Authors
Two principal points of criticism are warranted:
1. Since the rat brain has low content of white matter in its hemispheres, the primary area of ischemic necrosis in the middle cerebral artery occlusion model, the pathology initiated in this model is primarily that of the gray matter injury. This is of little relevance to human stroke where a white matter injury and subsequent very severe, destructive and extraordinarily protracted inflammation is initiated. A much larger animal brain with large content of the white matter would be required to serve as a proper animal model, a porcine brain for example. Alternatively, since the spinal cord is rich in the white matter, SCI in the rat should be considered a proper model of stroke.
2. From a medical perspective, emphasis on pathologic mechanisms of stroke occurring within the initial 24 hrs may be of less than highest importance in surviving patients. It is the severity, destructive power, prolonged nature and perilesional vasogenic edema that are initiated in the day 3 (judging from the systematic work on the rat model) and that can be inhibited with effective anti-inflammatory treatments. Efficacy of such treatments needs to be systematically monitored but blood tests are not yet available. Measuring levels of CNS lipids in systematic blood samples may at least partially address this demand.
Author Response
Reviewer #2:
We would like to express our sincere gratitude to the reviewer for assessing our manuscript and offering distinct means of improving it.
Two principal points of criticism are warranted:
- Since the rat brain has low content of white matter in its hemispheres, the primary area of ischemic necrosis in the middle cerebral artery occlusion model, the pathology initiated in this model is primarily that of the gray matter injury. This is of little relevance to human stroke where a white matter injury and subsequent very severe, destructive and extraordinarily protracted inflammation is initiated. A much larger animal brain with large content of the white matter would be required to serve as a proper animal model, a porcine brain for example. Alternatively, since the spinal cord is rich in the white matter, SCI in the rat should be considered a proper model of stroke.
- Although the middle cerebral artery occlusion (MCAO) model is a well-established ischemic stroke model in rodents, we acknowledge that the pathology and anatomical structure of the rat brain present certain limitations when drawing direct comparisons to human stroke. Consequently, we have made efforts to clarify that our findings are preliminary, specific to the rat model, and warrant further validation in models or species that more accurately reflect human clinical conditions.
- Lines 26: “Among five LPC species, plasma LPC(16:0) and LPC(18:1) showed strong correlations with sensorimotor dysfunction, lesion severity, and mechanistic biomarkers in the rat stroke model.”
- Lines 29-31: “This study advances the understanding of LPC as a blood marker in ischemic stroke and highlights directions for future research to further elucidate its association with stroke severity, particularly through investigations in more clinically representative models.”
- Lines 360-363: “Furthermore, interspecies variation and age-related risk factors emphasize the importance of examining the relationship between plasma LPC levels and stroke severity in more clinically relevant models to confirm these findings.”
- From a medical perspective, emphasis on pathologic mechanisms of stroke occurring within the initial 24 hrs may be of less than highest importance in surviving patients. It is the severity, destructive power, prolonged nature and perilesional vasogenic edema that are initiated in the day 3 (judging from the systematic work on the rat model) and that can be inhibited with effective anti-inflammatory treatments. Efficacy of such treatments needs to be systematically monitored but blood tests are not yet available. Measuring levels of CNS lipids in systematic blood samples may at least partially address this demand.
- We appreciate the reviewer's acknowledgment that measuring markers of central nervous system dyslipidemia in plasma could serve as a potential method for assessing ischemic stroke development. However, we also recognize the reviewer’s point that assessments conducted at 24 hours may miss critical time points most relevant for surviving ischemic stroke patients. Consequently, we have added a statement highlighting the need for further validation of these findings at a more sub-acute time point.
- Lines 337-340: “Furthermore, assessing LPC levels at a later time point, such as 72 hours post-stroke, could provide additional insights into the relationship between LPC and the prolonged degradative mechanisms associated with ischemic stroke, including inflammation and edema [76–78].”
Lines 341: “Secondly, stroke severity was assessed exclusively at the 24-hour mark; longer-term evaluations, extending over weeks to months, may more accurately reflect chronic outcomes, as clinical studies typically emphasize prolonged timeframes, often spanning several months [6,79,80].”

Round 2
Reviewer 1 Report
Comments and Suggestions for Authors
The authors have successfuly addressed my concerns, the manuscript can be published in its present form
Comments on the Quality of English LanguageEnglish, fine for this purpose
Author Response
The authors have successfuly addressed my concerns, the manuscript can be published in its present form.
- We thank the reviewer for taking the time to evaluate our manuscript and contributing to its improvement.

Reviewer 2 Report
Comments and Suggestions for Authors
The authors addressed the limitations of the experimental model offered by the rat brain in somewhat circumspect way which is a progress in the right direction but it could be further improved to enhance the value of the paper.
- a sentence in the introduction specifying limitations of the rat model of stroke due to paucity, not reduction(!, it is a normal amount of white matter in the rat brain) of the white matter, would be helpful.
- with a clear understanding that this study involves and acute model of brain injury not extending beyond 24 hours, I would recommend that in discussion authors consider the fate of a patient surviving stroke, available knowledge of the pathogenesis of neurotrauma in presence of large quantities of damaged myelin and introduce the need for blood tests including CNS-specific lipids to systematically monitor the severity of the disease initiated by stroke in patients. This will be soon (hopefully) badly needed to determine the success or failure of anti-neuroinflammatory treatments.
Author Response
The authors addressed the limitations of the experimental model offered by the rat brain in somewhat circumspect way which is a progress in the right direction but it could be further improved to enhance the value of the paper.
- a sentence in the introduction specifying limitations of the rat model of stroke due to paucity, not reduction(!, it is a normal amount of white matter in the rat brain) of the white matter, would be helpful.
- We appreciate the reviewer’s insightful suggestion and have adjusted the introduction accordingly.
- Lines 78-80: “a well-established pre-clinical model despite the anatomical differences in brain structure between rats and humans, particularly the lower white matter-to-gray mat-ter ratio in rats.”
- with a clear understanding that this study involves and acute model of brain injury not extending beyond 24 hours, I would recommend that in discussion authors consider the fate of a patient surviving stroke, available knowledge of the pathogenesis of neurotrauma in presence of large quantities of damaged myelin and introduce the need for blood tests including CNS-specific lipids to systematically monitor the severity of the disease initiated by stroke in patients. This will be soon (hopefully) badly needed to determine the success or failure of anti-neuroinflammatory treatments.
- We appreciate the reviewer's detailed feedback on the role of myelin breakdown in ischemic stroke recovery. In response, we have formulated the following statements to address the relevance of blood tests for lipid components after ischemic stroke, to the best of our ability. If further clarification is needed, we would welcome additional guidance.
- Lines 333-337: “Lipid dysregulation, neuroinflammation, and myelin breakdown are pivotal contributors to stroke pathogenesis, leading to cognitive decline and neurobehavioral dysfunction [13]. Therefore, the development of blood tests focusing on specific lipids, particularly LPC(16:0) and LPC(18:1), is crucial for enhancing prognostic accuracy and for evaluating clinical trial outcomes.”
